# Structural Design of Aerostatic Bearing Based on Multi-Objective Particle Swarm Optimization Algorithm

Biqing Ye, Guixin Yu , Yidong Zhang and Gang Li *

Key Laboratory of Special Purpose Equipment and Advanced Processing Technology, Ministry of Education, Zhejiang University of Technology, Hangzhou 310023, China
* Correspondence: ligang@zjut.edu.cn; Tel.: +86-139-5819-9396

**Abstract:** Aerostatic bearings are considered crucial components that can improve the measurement accuracy of ground simulation tests of space equipment. A structural optimization design method is proposed to enhance the static performance of aerostatic bearings. A mathematical model which can quickly calculate the aerostatic bearing capacity and gas consumption is established, and the influence of structural parameters on bearing performance is analyzed using simulation software. By comparing the convergence time and convergence results of the algorithm using different initialization methods, the Latin hypercube initialization method is selected instead of the random initialization method. The multi-objective particle swarm optimization algorithm is used to obtain the optimal solution set distributed in the objective space. It is found that the optimized structural parameters meet the requirements of improving the capacity and reducing gas consumption, which verifies the method's effectiveness in designing the structural parameters of aerostatic bearings.

**Keywords:** aerostatic bearing; bearing static performance; simulation analysis; improved multi-objective particle swarm optimization algorithm; structure design



## 1. Introduction

In space equipment, it is necessary to detect the output torque, stiffness, precision, and other indicators of the joint of the space manipulator in the ground simulation test to ensure the reliability of the operation [1,2]. Compared with ordinary bearings, gas bearings have the characteristics of extremely low friction resistance, clean media, and good adaptability to high-speed rotational environments [3]. They show great application prospects for improving the accuracy of the ground tests of space manipulator joints. Zhao et al. summarized the research developments of aerostatic bearings and pointed out that the applications of aerostatic bearings are affected by low load capacity and stability [4]. However, the development of high-speed machinery and high-precision machinery has demanded higher performance of aerostatic bearings. Therefore, a reliable optimization design method is needed to design the structural parameters of aerostatic bearings.

There are many structural parameters and physical parameters that affect the static performance of aerostatic bearings. In the design process, there is a competitive relationship between improving bearing capacity and reducing gas consumption. Therefore, optimizing parameters while considering the static performance of aerostatic bearings has become the key to the design of aerostatic bearings. Zhu et al. used the DIRECT and simulated annealing (SA) algorithms to design the air slider. They found that the DIRECT algorithm is more efficient in solving the non-linear slider optimization problem [5]. Wang et al. used the hypercube segmentation method instead of the genetic algorithm to optimize the air-bearing parameters. This algorithm shows the characteristics of a wide calculation range and a small calculation amount [6]. Shie et al. used a hybrid genetic optimization algorithm to calculate the optimal parameters of air bearing and verified the reliability of the optimization design using experiments [7]. Li et al. used the maximum Mach number

to predict bearing dynamic stability. Accordingly, they established the optimization model to improve stiffness and eliminate micro-vibration [8]. Cao et al. established two theoretical models of an ultra-low-friction cylinder to calculate the cylinder's performance. Then, they optimized the cylinder's structure using a constrained optimal design method [9]. The particle swarm optimization algorithm (PSO) is an evolutionary algorithm that simulates bird predation. It is simple to implement and has been successfully applied to multi-objective optimization problems [10]. Naderipour et al. converted two optimization objective functions into a single objective function using the linear weighting method. Then, they used the PSO algorithm to find the optimal combination of design variables [11]. Zhu et al. proposed an external archive-guided multi-objective particle swarm optimization algorithm (AgMOPSO) and introduced the update mechanism of the archive. They verified that AgMOPSO effectively improves the convergence speed when dealing with multi-objective test problems [12]. Chen et al. proposed a double-loop multi-objective particle swarm optimization algorithm to solve the problem when there are uncertain variables in the optimization process [13]. Lee used MOPSO to optimize the structural parameters of the turbofan engine and verified the convergence of MOPSO by comparing the Pareto front with four solutions obtained using the single-objective optimization algorithm [14]. Gao et al. improved the distribution of sub-problems using a weight vector adjustment strategy instead of Euclidean distance to divide congestion degrees [15]. Jing et al. used a coupled multi-objective optimization algorithm to optimize the screw centrifugal blood pump and the Latin hypercube sampling method to generate an initial sample with a more uniform distribution [16]. To save computation costs and ensure the reliability of the design method, this paper uses the Latin hypercube initialization method instead of random initialization. Then, the optimization results using different initialization methods are analyzed.

Another key to applying the multi-objective particle swarm optimization algorithm to the optimal design of aerostatic bearings is calculating the static performance (bearing capacity and gas consumption) through the basic structural parameters of the bearings. The Reynolds equation of compressible fluid is the basic equation for calculating the performance of gas bearings. Therefore, the essence of calculating the static performance of aerostatic bearings is to solve the Reynolds equation. Mori researched the pressure distribution of the gas-lubricated circular thrust bearing by assuming that the flow pattern near an inlet could be divided into the supersonic flow region, the subsonic flow region, and the viscous and isothermal flow region. He found that the pressure distribution at the orifice outlet was not consistent with the Reynolds equation [17]. Lund used the spring and damping coefficients to study the dynamic performance and stability of gas bearings [18]. Majumdar solved the continuity equation and the Reynolds equation using the finite difference method to analyze the steady-state characteristics of porous gas bearings [19]. Professor Liu et al. simplified the two-dimensional Reynolds equation into a one-dimensional one using the linear gas source assumption, which greatly shortened the calculation time of the static characteristics of aerostatic bearings compared with the numerical calculation method [20]. Malik and Bert applied the differential quadrature method to solve steady-state oil and gas lubrication problems in self-acting hydrodynamic bearings [21]. Lo et al. derived the non-linear dimensionless Reynolds equation of the gas flow within bearings and discretized it using the Newton method [22]. Yoshimoto et al. analyzed the pressure distribution in the bearing clearance of circular aerostatic thrust bearings with a single air supply inlet using CFD technology. It was found that the airflow state will change in the rapid pressure recovery region [23]. Yang et al. used the Ph-method to analyze the Reynolds equation and solved it using the finite difference method and numerical integration. They analyzed the influence of the number and locations of entry holes on the characteristics of aerostatic bearings [24]. Miyatake et al. determined discharge coefficients for a small feed hole of an aerostatic thrust bearing by comparing the results of a CFD simulation and a mathematical calculation model. They investigated the static and dynamic characteristics of aerostatic thrust bearings using FDM to solve the Reynolds equation and obtain the pressure distribution. It was found that the orifice could improve

the damping and stiffness coefficient [25]. Zhao et al. used the finite difference method (FDM) to analyze the pressure distribution of the multi-supply air cushion. According to the principle of mass conservation, the correction iteration route of the pressure boundary conditions of each supply hole was developed, and the pressure distribution and gas flow under eccentric load were calculated [26]. Qiang et al. used the linear perturbation method to solve the Reynolds equation. They found that the bearing capacity and the critical mass of hydrogen gas bearings were slightly lower than other lubricating mediums [27].

Although the numerical calculation method shows high accuracy, there are some problems, such as a large amount of calculation, complex calculation steps, and a large amount of result data. Therefore, it is unsuitable as a solution method for the objective function in the iterative process of the optimization algorithm. To decrease the calculation time of the static performance, the engineering calculation model is established. In addition, this paper also establishes a simulation model of an aerostatic bearing to verify the result of the engineering calculation model and analyze the influence of the structural parameters of the aerostatic bearing on static performance. Then, the multi-objective particle swarm optimization algorithm is used to optimize the structural parameters of the aerostatic bearing. Finally, the simulation model verifies the reliability of the optimization results, confirming the feasibility of rapid optimization design.

## 2. Mathematical Model of Aerostatic Bearing

### 2.1. Structure of Aerostatic Bearing

The structure of the aerostatic bearing is shown in Figure 1. Aerostatic bearing adopts double row orifice structure; the key parameters of the structure are bearing length $L$, journal diameter $D$, distance from the center of the orifice to the outlet $l$, orifice diameter $d$, gas film thickness $h$, eccentricity $e$, gas supply pressure $P_s$, and atmospheric pressure $P_a$. When the journal is not eccentric, the gas film thickness is consistent at any circumferential position, which is recorded as the average gas film thickness $h_{av}$.

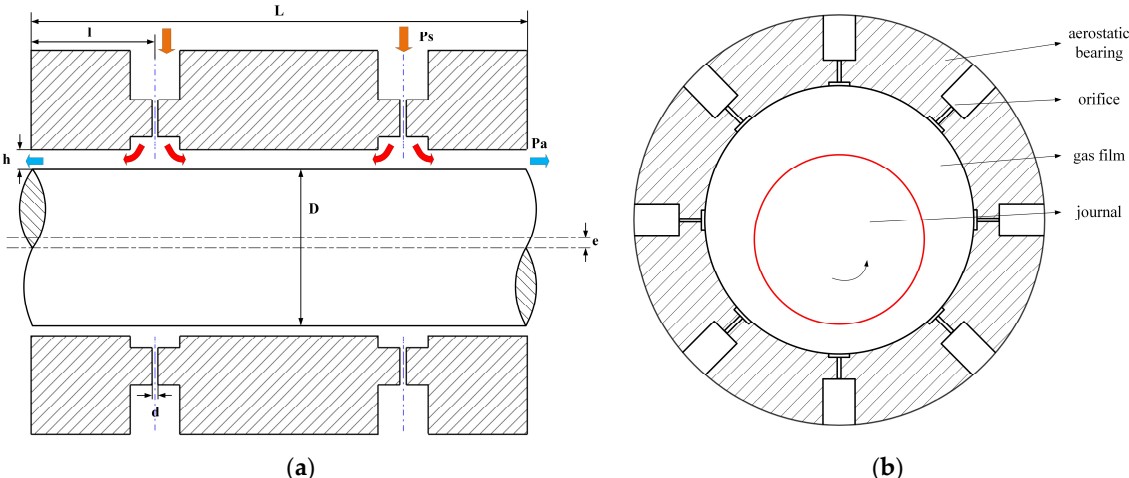

**Figure 1.** (**a**) Structure diagram of aerostatic bearing; (**b**) orifice distribution diagram.

### 2.2. Engineering Calculation Method

The engineering calculation method usually expands the gas film of the aerostatic bearing into a plane for analysis. According to the number of single row orifice $n$, it is divided into $n$ equal parts; each width is $2\pi R/n$, and the following assumptions are made:

1.  The gas film thickness is very small relative to the diameter of the journal. The curvature of the gas film can be ignored. It is considered that the thickness of the gas film in each equal area is a fixed value;
2.  The gas film thickness is tens of microns. It can be considered that the gas pressure and density distribution in the direction of the gas film thickness shows no change;

3.    Lubricating gas does not slide on the solid surface, and the velocity of gas on the solid surface is the same as that on the solid surface;
4.    As the lubricating gas is air and the density is small, the effect of volume force can be ignored, and the gas flow in the lubrication process is assumed to be laminar flow;
5.    It is considered that the pressure between the two orifices in each equipartition area is equal, and the pressure is equal to the pressure after gas throttling. It is assumed that there is no gas flow between each equipartition area.

The gas film expansion diagram is shown in Figure 2. The direction of the gas flowing to the end face is the $x$-axis, the direction of the gas film thickness is the $y$-axis, and the circumferential direction is the $z$-axis. The gas film thickness at the orifice represents the gas film thickness in an equal area. According to the assumption that the gas in each equal area shows only one-dimensional motion flowing axially to the bearing outlet, the gas momentum equation can be expressed as:

$$\begin{cases} \frac{\partial P}{\partial x} = \mu \frac{\partial^2 u}{\partial y^2} \\ \frac{\partial P}{\partial y} = 0 \\ \frac{\partial P}{\partial z} = 0 \end{cases} \tag{1}$$

The gas mass flow $m_{i1}$ of gas flowing out of the $i$th region can be obtained using Equation (2):

$$m_{i1} = b \cdot m_{ix} = \frac{2\pi R}{n} \int_0^{h_i} \rho u \, dy \tag{2}$$

where $m_{ix}$ is the mass flow per unit width section perpendicular to the $x$-axis.

The gas film thickness $h_i$ of the $i$th region can be obtained using Equation (3):

$$h_i = h_{av}(1 - \varepsilon \cos \varphi_i) \tag{3}$$

where $h_{av}$ is the average gas film thickness; $\varepsilon$ is eccentricity; and $\varphi_i$ is the position of the orifice on the circumference of the bearing.

The gas mass flow $m_{i2}$ of gas flowing into the $i$th region can be obtained using Equation (4):

$$m_{i2} = A P_s \phi \sqrt{\frac{2\rho_s}{P_s}} \psi_i \tag{4}$$

where $A$ is the cross-sectional area of the orifice; $\phi$ is the gas flow coefficient, and its value is 0.8; and $\psi_i$ is the flow function of the orifice, which is determined by the air supply pressure and orifice outlet pressure.

$$\psi_i = \begin{cases} \left[ \frac{k}{2}\left(\frac{2}{k+1}\right)^{(k+1)/(k-1)} \right]^{1/2} &, \left( \frac{P_{0i}}{P_s} \le \beta_k \right) \\ \left\{ \frac{k}{k-1}\left[ \left(\frac{P_{0i}}{P_s}\right)^{2/k} - \left(\frac{P_{0i}}{P_s}\right)^{(k+1)/k} \right] \right\}^{1/2} &, \left( \frac{P_{0i}}{P_s} > \beta_k \right) \end{cases} \tag{5}$$

where $P_s$ and $P_{0i}$ are the gas supply pressure and the orifice outlet pressure of the $i$th region, respectively, and $\beta_k$ is the critical pressure ratio; the orifice throttling process can be regarded as an adiabatic process, where $k$ is the adiabatic index, with a value of 1.4, so the critical pressure ratio is 0.528.

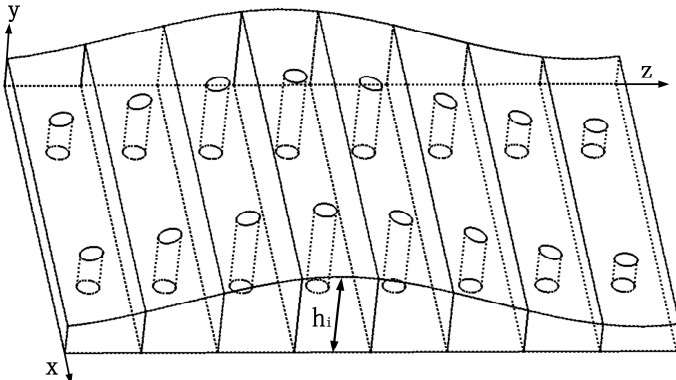

**Figure 2.** Sketch of gas film in the circumferential direction.

The expression of the inflow gas mass flow in each region can be obtained by integrating Equation (1) in the direction of gas film thickness and bringing it into Equation (2):

$$m_{i1} = -\frac{\pi R h_i^3 \rho_a}{6\mu n P_a} P \frac{dP}{dx} \tag{6}$$

By separating variables and integrating $x$ and $P$, we can obtain:

$$P^2 - P_{0i}^2 = -\frac{12\mu n P_a}{\pi R h_i^3 \rho_a} m_{i1} \cdot x (0 \le x \le l) \tag{7}$$

Inserting the boundary condition when $x = l$, $P = P_a$, we can obtain:

$$P_a^2 - P_{0i}^2 = -\frac{12\mu n P_a}{\pi R h_i^3 \rho_a} m_{i1} l \tag{8}$$

It can be seen from the continuity equation that $m_{i1} = m_{i2}$. We can bring $m_{i2}$ into Equation (8) and simplify it according to the gas state equation to obtain:

$$P_{0i}^2 - P_a^2 = \frac{12\mu n}{\pi R h_i^3} l A \phi P_s \sqrt{\frac{2P_a}{\rho_a}} \cdot \psi_i \tag{9}$$

The projection of the carrying capacity $F_i$ provided by the air film in the $i$th region on the vertical direction can be obtained by:

$$F_i = 2 \left[ R P_{0i}(L - 2l) \sin \frac{\pi}{n} + \int_0^l 2RP \sin \frac{\pi}{n} dx \right] \cos \alpha_i \tag{10}$$

Thus, the total carrying capacity $W$ can be obtained by superimposing the carrying capacity provided by the gas film of each region:

$$W = \sum_{i=1}^n F_i = 2R \sin \frac{\pi}{n} \sum_{i=1}^n \left[ P_{0i}(L - 2l) + \int_0^l P dx \right] \cos \alpha_i \tag{11}$$

The expression of pressure $P$ can be obtained from Equations (7) and (8). Then, we can bring it into Equation (11):

$$W = 2R \sin \frac{\pi}{n} \sum_{i=1}^n P_{0i} \left[ (L - 2l) + \frac{4}{3} l \frac{1 - \left( \frac{P_a}{P_{0i}} \right)^3}{1 - \left( \frac{P_a}{P_{0i}} \right)^2} \right] \cos \alpha_i \tag{12}$$

The orifice outlet pressure of the $i$th region $P_{0i}$ can be obtained by solving the nonlinear Equation (9) numerically. Then, the gas consumption of each region can be obtained using Equation (13):

$$Q = AP_s\phi\sqrt{\frac{2\rho_s}{P_s}}\sum_{i=1}^{n}\psi_i \tag{13}$$

## 3. Simulation Analysis

### 3.1. Simulation Modeling of Aerostatic Bearing

Fluent can accurately solve the pressure distribution of the gas-bearing flow field using the finite volume method. The Fluent simulation process is shown in Figure 3.

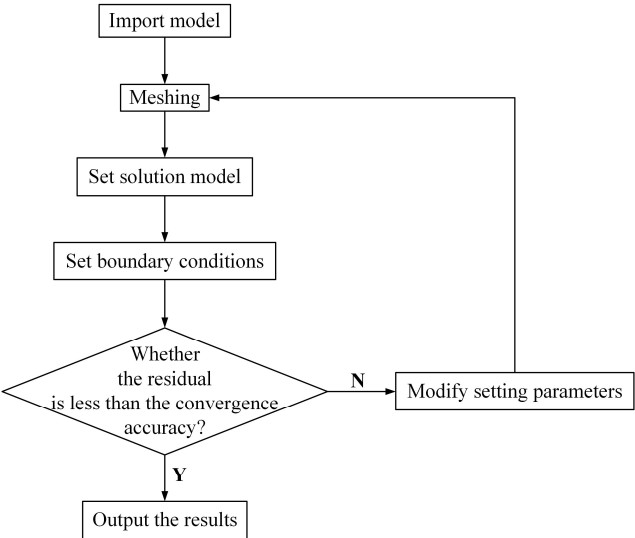

**Figure 3.** Simulation process.

There are two rows of orifices symmetrically arranged on the aerostatic bearing. Therefore, the 1/2 geometric model of the flow field of the gas bearing is established in SolidWorks. The flow field mesh is shown in Figure 4.

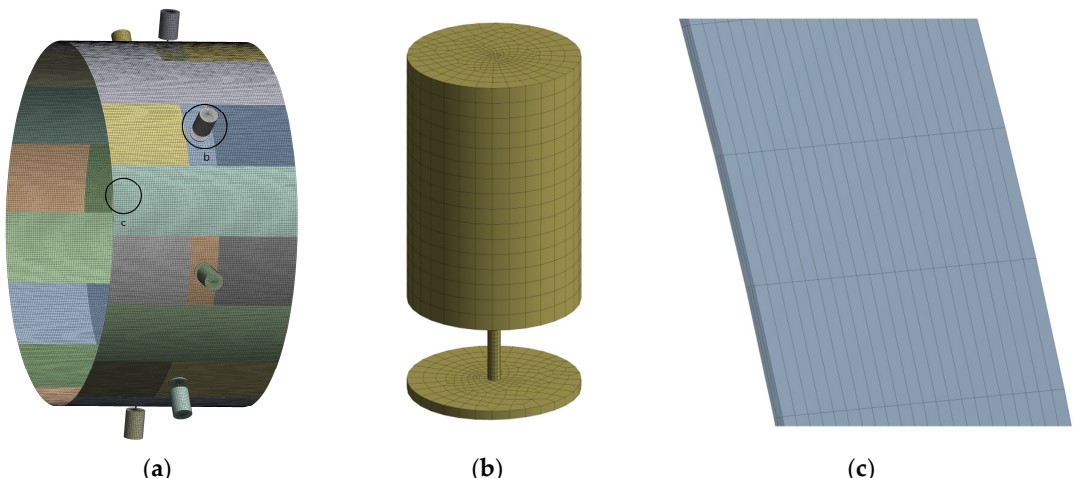

(**a**)  (**b**)  (**c**)

**Figure 4.** (**a**) Gas film flow filed meshing (1/2 model); (**b**) orifice meshing densification; (**c**) meshing layering.

Due to the large difference in the order of magnitude between the gas film thickness and other structural parameters and to ensure the reliability of the solution results, the mesh is divided into three layers in the direction of the gas film thickness, and the mesh

at the orifice is encrypted. As the gas flow is assumed to be laminar in the engineering calculation, the laminar model is used, and the solver is set to the pressure basis to solve and open the energy equation. After the solution is completed, the static performance of the aerostatic bearing can be extracted in the post-processing module CFD-Post.

### 3.2. Engineering Calculation Model and Simulation Model Validation

In Section 3.5 of the literature [28], the author compares the theoretical calculation value and the simulation value of the bearing capacity of the aerostatic bearing with two sets of experimental detection values. It is found that the theoretical calculation value and the simulation value are well fitted. The simulation value is consistent with the experimental test results and can be used as a theoretical analysis model. Further, the author provides the detailed process and calculation results of the theoretical calculation of the bearing capacity of the aerostatic bearing in the literature [29]. We bring the structural parameters provided by the literature in [29] into the engineering calculation and simulation models established in this paper to compare the results.

It can be seen from Figure 5 that the calculation results of the bearing capacity obtained using the engineering calculation model and the simulation model established in this paper are in good agreement with the calculation results presented in the literature [29]. The maximum error between the engineering calculation value and the literature result is 2.7% when the eccentricity is 0.3. The maximum error between the simulation value and the literature result is 6.2% when the eccentricity is 0.1. Under the same eccentricity, the calculated bearing capacity of the theoretical solution is always greater than the simulation result. As the diffusion and circulation effects are not considered in the theoretical calculation, the calculated bearing capacity will be larger than the real value. However, the overall error is small, so the engineering calculation results and simulation results are acceptable. Based on this simulation model, the optimization range is determined, and the effectiveness of the optimization results is tested. The structural parameter optimization algorithm is designed based on this engineering calculation model.

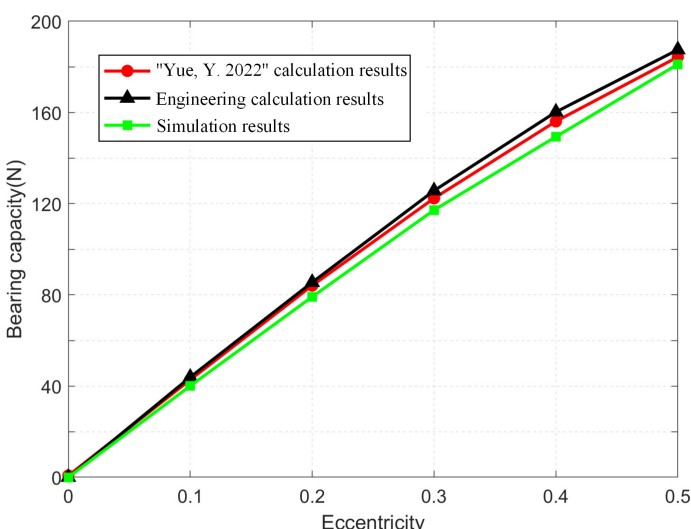

**Figure 5.** Relationship between bearing capacity and eccentricity [29].

### 3.3. Optimization Parameter Analysis

Many factors affect the static performance of the aerostatic bearing. Air supply pressure $P_s$, eccentricity $\varepsilon$, bearing length $L$, and journal diameter $D$ are determined using real working conditions, so they can be set as fixed values in the design process. The influence of orifice diameter $d$, the distance from the orifice to the outlet $l$, the average gas film thickness $h_{av}$, and the number of single-row orifice $n$ on the static performance

of the aerostatic bearing can be studied using Fluent, and the optimization range can be determined. The initial values of the structural parameters are shown in Table 1.

**Table 1.** Structural parameters of aerostatic bearing.

| Structural Parameters | Initial Value | Unit |
|---|---|---|
| Air supply pressure, $P_s$ | 0.4 | MPa |
| Bearing length, $L$ | 60 | mm |
| Journal diameter, $D$ | 30 | mm |
| Orifice diameter, $d$ | 0.2 | mm |
| Orifice to outlet distance, $l$ | 15 | mm |
| Average gas film thickness, $h_{av}$ | 20 | μm |
| Equalizing chamber diameter, $d_a$ | 2 | mm |
| Equalizing chamber depth, $h_a$ | 0.1 | mm |
| Single-row orifices number, $n$ | 8 | / |

### 3.3.1. Influence of Orifice Diameter on Static Performance

It can be seen from Figure 6a that the diameter of the orifice has little effect on the bearing capacity at low eccentricity. Under the same eccentricity, the increase in the diameter of the orifice leads to the deterioration of the throttling effect, which reduces the overall bearing capacity. As the gas consumption is proportional to the square of the orifice diameter, the increase in the orifice diameter will lead to a significant increase in gas consumption, consistent with the results in Figure 6b.

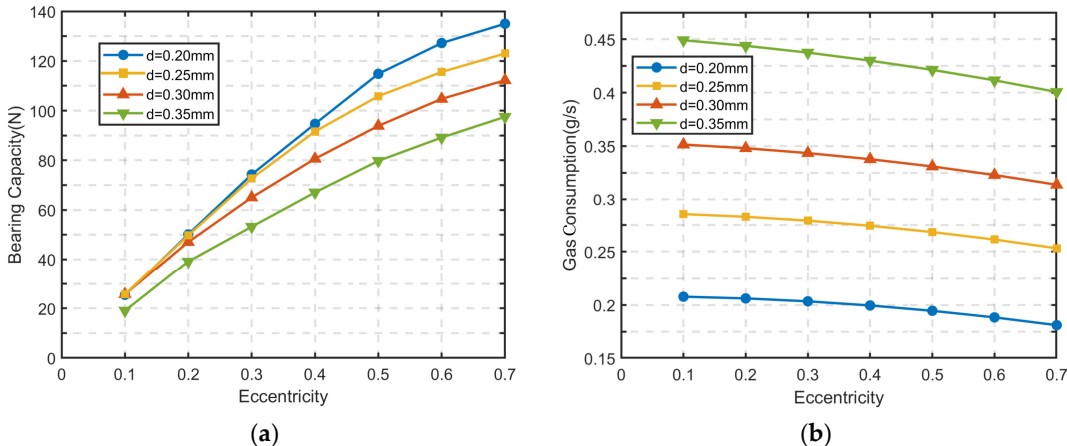

**Figure 6.** Influence of orifice diameter on (**a**) bearing capacity and (**b**) gas consumption.

### 3.3.2. Influence of Orifice to Outlet Distance on Static Performance

It can be seen from Figure 7a that the bearing capacity under the different conditions changes almost uniformly with the eccentricity. When the orifice is closer to the outlet, the deceleration phase of the compressed gas flowing into the atmosphere after throttling is shorter, resulting in increased gas consumption, as shown in Figure 7b.

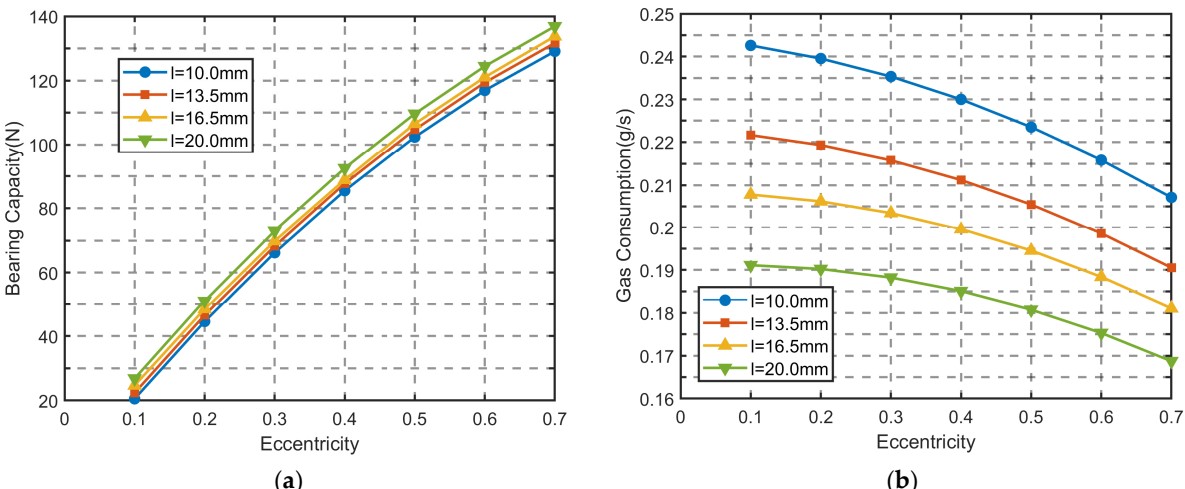

**Figure 7.** Influence of orifice position on (**a**) bearing capacity and (**b**) gas consumption.

### 3.3.3. Influence of Average Gas Film Thickness on Static Performance

When the gas film is thinner, the pressure dissipation of the compressed gas after throttling is smaller. Therefore, as shown in Figure 8a, at the same eccentricity, reducing the film thickness can improve bearing capacity and reduce gas consumption. However, it should also be noted that smaller film thicknesses will challenge the difficulty of manufacturing and assembly.

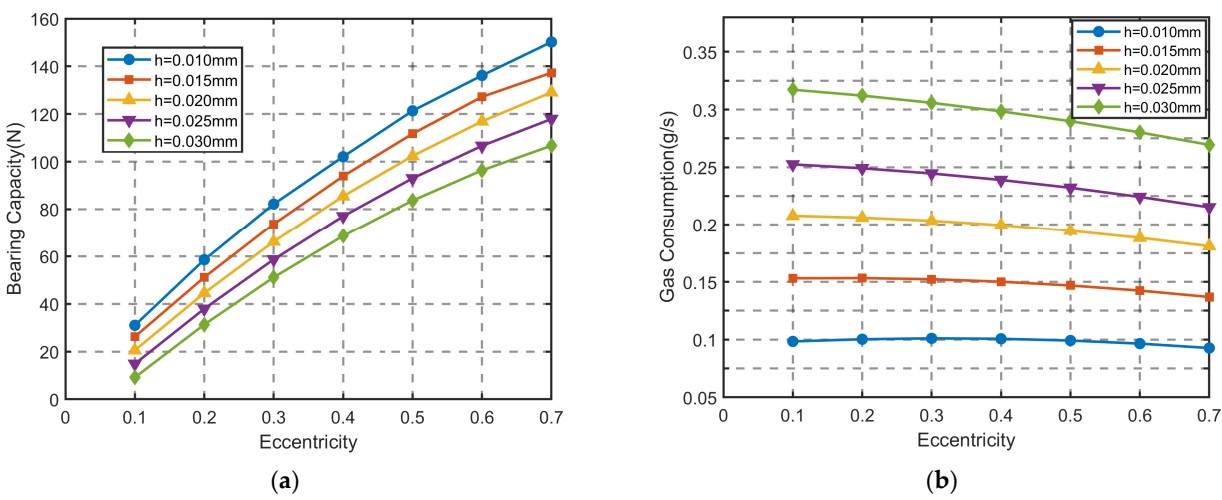

**Figure 8.** Influence of average film thickness on (**a**) bearing capacity and (**b**) gas consumption.

### 3.3.4. Influence of Orifice Number of Each Row on Static Performance

As shown in Figure 9a, under the same eccentricity, the increase of orifices can provide greater bearing capacity and improve the stiffness of the gas bearing but will inevitably increase gas consumption.

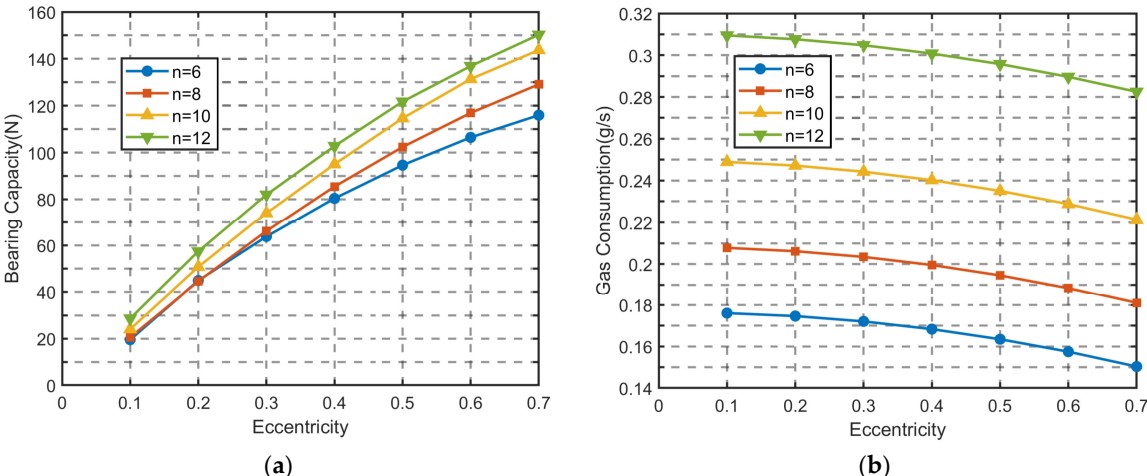

**Figure 9.** Influence of the number of orifices on (**a**) bearing capacity and (**b**) gas consumption.

## 4. Optimization Design Algorithm

### 4.1. Optimization Model

According to the analysis of the simulation results, the optimization range of the structural parameters to be optimized is determined. The optimization range is shown in Table 2. The other structural parameters are fixed to the initial value set in Section 3.3, and the eccentricity of the gas bearing is fixed to 0.4 during the optimization process. The optimization goal is to improve the bearing capacity while minimizing gas consumption.

**Table 2.** Optimization range of structural parameters.

| Structural Parameters | $d$ (mm) | $l$ (mm) | $h_{av}$ (mm) | $n$ |
|---|---|---|---|---|
| Initial value | 0.2 | 15 | 0.02 | 8 |
| Optimization range | 0.2–0.3 | 10–20 | 0.015–0.025 | 6–10 |

According to the derivation of bearing capacity and gas consumption in the engineering calculation method, the following optimization model can be obtained:

$$\min F(X) = [f_1(X) = -W, f_2(X) = Q]$$

$$s.t. \quad x_i^{\min} \leq x_i \leq x_i^{\max}, i = 1, 2, 3, 4$$

$$X = [x_1, x_2, x_3, x_4] = [d, l, h_{av}, n]$$

where $f_1$ is the objective function of the bearing capacity, and $f_2$ is the objective function of gas consumption. The $X$ vector consists of four structural parameters, and the boundary condition is the optimization range of the four structural parameters.

### 4.2. Ordinary Multi-Objective Particle Swarm Optimization Algorithm

The ordinary multi-objective particle swarm optimization algorithm can be split into a basic particle swarm optimization algorithm and an archives maintenance algorithm.

#### 4.2.1. Basic Particle Swarm Optimization Algorithm

In the particle swarm optimization algorithm, each particle is denoted by $x_i = [x_{i1}, x_{i2}, \ldots\ldots, x_{id}]$, where $i = 1, 2, \ldots\ldots, N$, $N$ is the number of particles, and $d$ is the dimension of the particle. Each particle updates the position using Equations (14) and (15) according to its optimal position and the optimal global position during the iteration process [30]:

$$v_i^{t+1} = w^t v_i^t + c_1 r_1^t \left(pb_i^t - x_i^t\right) + c_1 r_2^t \left(gb^t - x_i^t\right) \tag{14}$$

$$x_i^{t+1} = x_i^t + v_i^{t+1} \tag{15}$$

where $t$ is the number of iterations, $v_i^t$, $x_i^t$, and $pb_i^t$ are the speed, position, and self-optimal position of the $i$th particle at the $t$th iteration, $gb^t$ is the optimal global position at the $t$th iteration, $w^t$ is the inertia factor, $w^t$ decreases with the increase in the number of iterations, $c_1$ and $c_2$ are learning factors, and $r_1$ and $r_2$ are random numbers between 0 and 1.

### 4.2.2. Archives Maintenance Algorithm

In the process of multi-objective optimization, multiple Pareto solutions (non-dominated solutions) will be obtained. The solution set composed of all Pareto solutions is called the Pareto front (PF). Therefore, it is necessary to introduce an archive to store the Pareto solutions. The archive maintenance algorithm is as follows:

1. The target space is divided according to the boundary value of each objective function of all particles in the archive and the grid width;
2. Calculate the grid number of each particle in the archive;
3. Calculate the crowding degree of the grid, that is, the number of particles in the grid;
4. Determine whether the number of particles in the archive exceeds the archive's capacity. If it exceeds, randomly delete a particle in the most crowded grid until the number of particles does not exceed the archive's capacity.

### 4.2.3. Multi-Objective Particle Swarm Optimization

The iterative process of the multi-objective particle swarm optimization algorithm is as follows:

1. Randomly initialize each particle's position and velocity, calculate each particle's objective function value, update the optimal position of each particle, non-dominated sort each particle, save the Pareto solution to the archive, and call the archive maintenance algorithm;
2. Update the speed of each particle to obtain $v_i^{t+1}$ and then update the position of each particle to obtain $x_i^{t+1}$;
3. Calculate the objective function value of each particle after updating, and update the optimal position of each particle;
4. Non-dominated sort each particle, store the Pareto solution in the archive, and call the archive maintenance algorithm;
5. Determine whether the maximum number of iterations has been reached, jump to Step 2 if not, and output all Pareto solutions in the archive if reached.

### 4.3. The Latin Hypercube Initialization Method

The operation of the Latin hypercube initialization method is as follows:

1. Determine the population size $N$, which is the number of particles;
2. Divide the optimization range of each dimensional variable into $N$ equal parts. Randomly select a point in each interval of each dimension as a sampling point;
3. The sample points selected in each dimension space are randomly sorted to form the initial position of particles.

To illustrate the difference between the Latin hypercube initialization and the random initialization, the distributions of two-dimensional particles using different initialization methods are shown in Figure 10. It is found that the distribution of particles using Latin hypercube initialization is more uniform than those using random initialization, which is more conducive to algorithm optimization.

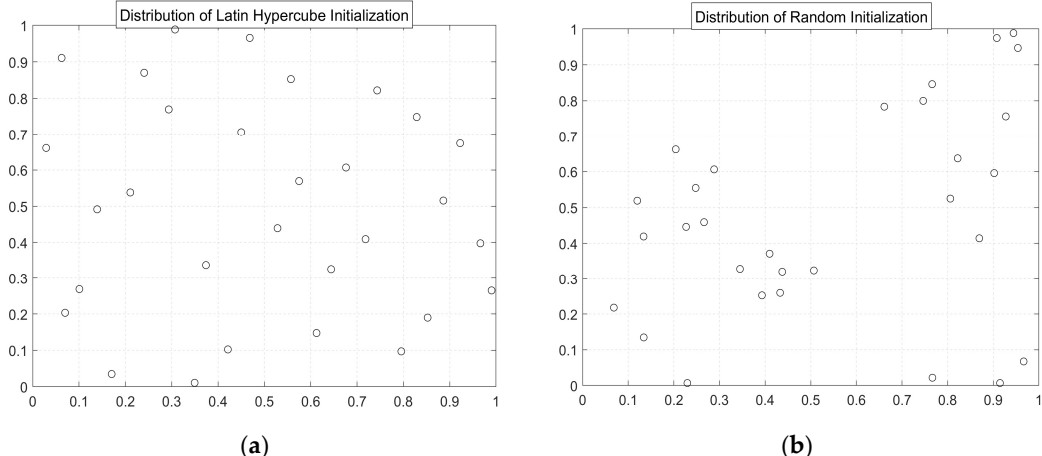

**Figure 10.** Distribution of (**a**) Latin hypercube and (**b**) random initialization.

### 4.4. Convergence Condition

The process of judging the convergence of the algorithm is as follows:

1.  Firstly, a large iteration period is preset as the stopping condition of the algorithm. Then, the objective function values corresponding to the Pareto solutions obtained by each iteration are stored. These objective function values are arranged from small to large according to the gas consumption (or according to the bearing capacity) to form a $2 \times n$ matrix $F_t$; $n$ is the number of Pareto solutions, and $t$ is the number of iterations;

2.  Determine whether the number of Pareto solutions of the $t + 1$ generation is the same as that of the previous generation. If it is different, the Pareto solutions are still changing, so let the algorithm continue to iterate. If not, go to the next step to judge its convergence;

3.  Equation (16) can express the convergence judgment formula. Firstly, the Frobenius norm of the difference between the $t + 1$ generation objective function matrix $F_{t+1}$ and the previous generation objective function matrix $F_t$ is calculated and then divided by the Frobenius norm of $F_t$. When the result is less than the given small amount $\varepsilon$, it is considered that the objective function matrix is almost unchanged, so the algorithm is considered to have converged:

$$\frac{\|F_{t+1} - F_t\|_F}{\|F_t\|_F} \leq \varepsilon \tag{16}$$

The algorithm's parameters, such as population number and iteration period, are determined through the trial and error method. We fixed the population number to 30, and the archive's capacity is 50. We compared the convergence of Latin hypercube initialization and random initialization and ran each algorithm ten times, respectively. The convergence situation is as follows (Table 3):

**Table 3.** Comparison of convergence period.

| Initialization Method | Average Convergence Period | Minimum Convergence Period | Maximum Convergence Period |
| --- | --- | --- | --- |
| Latin hypercube | 41.5 | 40 | 43 |
| Random | 47.2 | 39 | 55 |

Latin hypercube sampling can obtain a more diverse initial population distribution with less population size, and the convergence is very stable in 10 runs. The population initialization distribution using random sampling may be evenly filled in the target space

and too concentrated. Therefore, in 10 runs, the convergence period of random initialization fluctuates greatly. In the trial operation process, we obtained the maximum convergence period of 55, so we set the iteration termination condition to a maximum of 60 iterations to ensure the reliability of the algorithm's convergence in the real operation.

### 4.5. Results

#### 4.5.1. Comparison of Different Optimization Results

The different optimization results are illustrated in two aspects. Firstly, to compare the optimization results under different initialization methods, the progress of iteration is displayed in Figures 11 and 12, respectively. Secondly, the genetic algorithm (GA) is often used to solve multi-objective optimization problems. Therefore, to prove the effectiveness of particle swarm optimization, the performance of both algorithms in solving the same multi-objective optimization problem is compared. Identically, the population size of GA is 30 and is allowed to evolve to 60 generations. In the subsequent figures, the bearing capacity and gas consumption of the bearing are normalized.

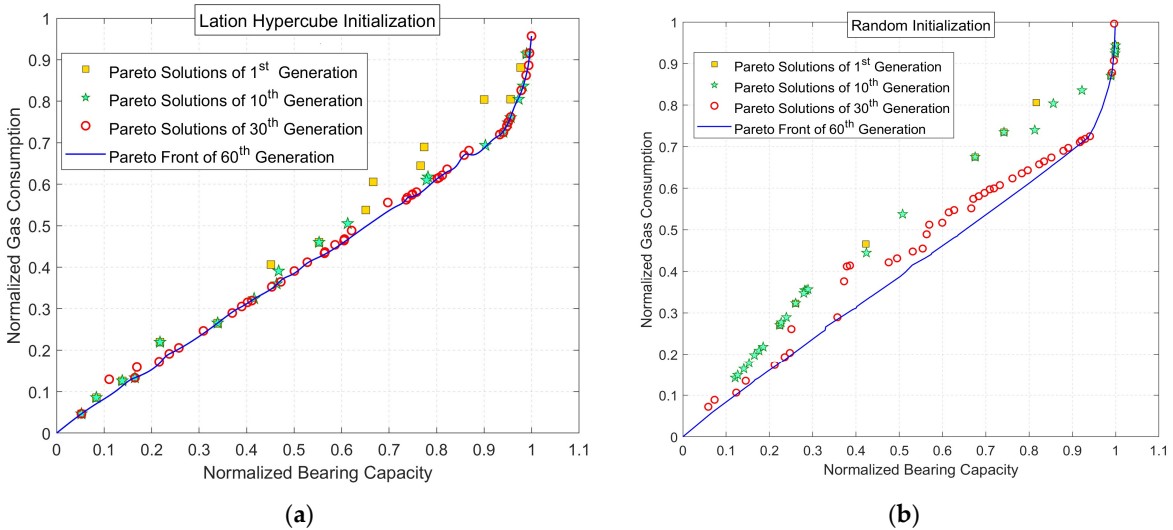

**Figure 11.** Pareto solutions obtained using (**a**) Latin hypercube and (**b**) random initialization.

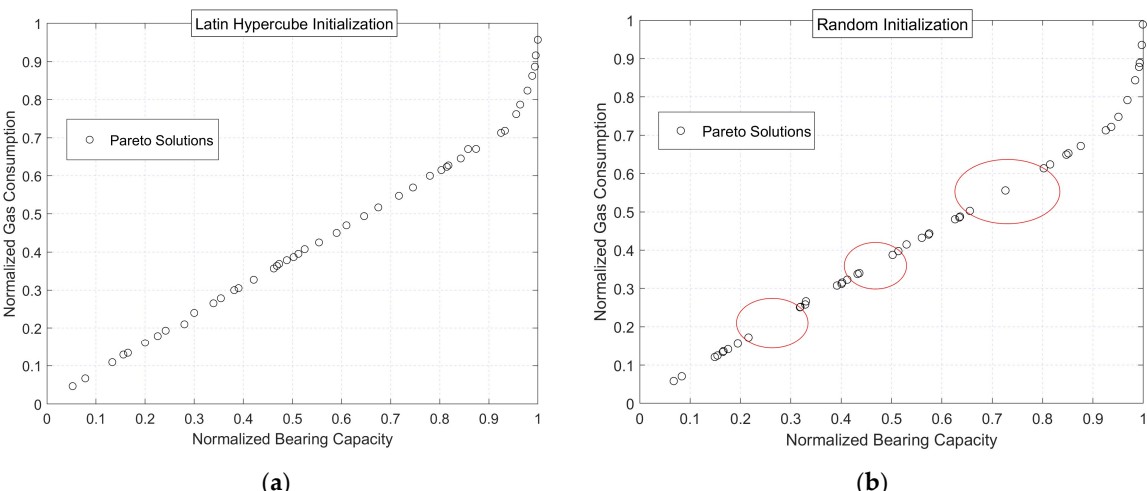

**Figure 12.** Pareto solutions obtained using (**a**) Latin hypercube and (**b**) random initialization after the 60th generation.

The ordinate corresponds to the normalized gas consumption, and the abscissa corresponds to the normalized bearing capacity. It can be seen from Figure 11a that the selectivity

of the Pareto solutions gradually increases from the 1st to the 30th generation. The particle swarm gradually approaches the Pareto front in the iterative process, and the Pareto solutions obtained after the 30th generation fit well with the final Pareto front. Compared with Figure 11b, it is found that the Pareto solutions obtained using random initialization after the 30th generation are still poorly fitted with the final Pareto front. As the particle distribution obtained by the Latin hypercube is more reasonable, it effectively reduces the amount of computation and improves the speed of finding the optimal solutions.

Figure 12 shows the Pareto solutions obtained using different initialization methods after the 60th generation. It can be seen from Figure 12b that the distribution of the Pareto solutions in the region identified by the red circle is sparse. Compared with Figure 12a, it is found that the Pareto solutions obtained using Latin hypercube initialization show a much wider distribution and more selections. It also proves that using Latin hypercube as a particle initialization method can effectively improve the global search ability.

It can be seen from Figure 13 that the Pareto solutions obtained using the two initialization methods after the 60th generation are in good agreement. The Pareto solutions obtained using GA are surrounded by the Pareto front obtained using particle swarm optimization in Latin hypercube initialization, which proves that particle swarm optimization shows a stronger global search ability and can find better solutions.

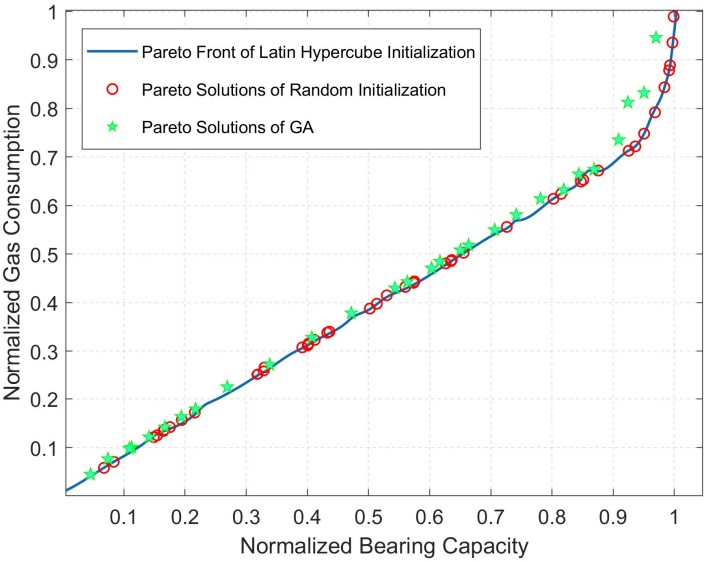

**Figure 13.** Pareto solutions obtained after the 60th generation.

The performance of different algorithms in solving the multi-objective optimization problem can be assessed using performance metrics, such as the spacing and diversity metric (DM) [31]. The performance metrics can be obtained using Equations (17) and (18), respectively. Spacing assesses the evenness of the solution set, and DM measures the diversity of the Pareto solutions. A more uniform and diverse solution set results in smaller values of spacing and DM. It can be seen from Figure 14 that particle swarm optimization using Latin hypercube initialization shows a better performance in solving the multi-objective optimization problem:

$$Spacing(A) = \sqrt{\frac{1}{A-1}\sum_{i=1}^{A}\left(\overline{d}-d_i\right)^2} \tag{17}$$

$$DM = \frac{d_f + d_l + \sum\limits_{i=1}^{A-1}\left|d_i - \overline{d}\right|}{d_f + d_l + (A-1)\overline{d}} \tag{18}$$

where $A$ is the number of Pareto solutions, $d_i$ is the Euclidean distance between consecutive solutions, $\bar{d}$ is the average of all distances, $d_i$, $d_f$, and $d_l$ are the Euclidean distances between the extreme solutions and the boundary solutions. In this paper, the intersection of the fitted Pareto front and the boundary of the target space are taken as the extreme solutions.

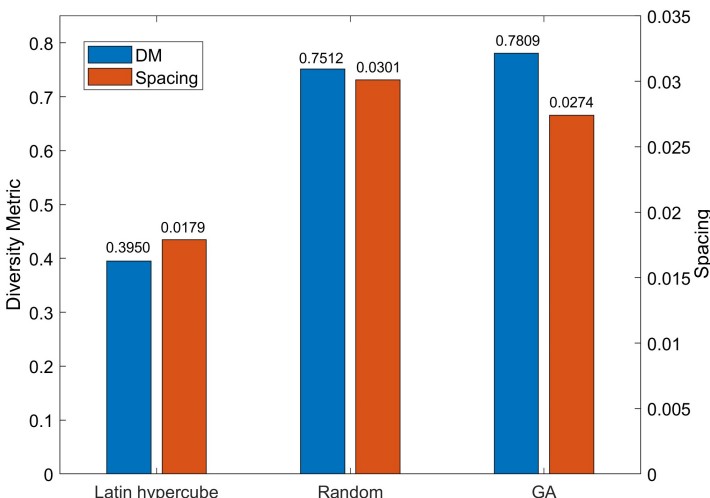

**Figure 14.** Performance metrics of the multi-objective optimization algorithm.

### 4.5.2. Structural Parameter Optimization Results

To further verify the effectiveness of the optimization solutions shown in Figure 12a, two groups of solutions with the minimum and maximum bearing capacity are selected from 50 groups of Pareto solutions, and eight groups of solutions are randomly selected from the remaining 48 groups of Pareto solutions according to the bearing capacity from small to large. The static performance corresponding to these ten groups of solutions is calculated using the simulation model and then compared with that of the aerostatic bearing before optimization. The comparison results are shown in Table 4.

Group 1 is the structural parameters before optimization, and Group 2 to Group 11 correspond to the ten groups of structural parameters obtained using the optimization algorithm. It can be seen intuitively from Table 4 that compared with the structural parameters before optimization, the ten groups of parameters optimized by the optimization algorithm guarantee the effect of improving bearing capacity and reducing gas consumption. Group 11 shows the largest increase in bearing capacity. It increases bearing capacity by 23.07% and reduces gas consumption by 1%. Group 2 shows the largest reduction in air consumption. It reduces gas consumption by 25.59% and increases bearing capacity by 2.96%. Therefore, it can be proved that the multi-objective particle swarm optimization algorithm can effectively optimize the structural parameters of aerostatic bearings, thereby improving the static performance.

**Table 4.** Comparison of optimization results.

| Group | $d$ (mm) | $l$ (mm) | $h_{av}$ (µm) | $n$ | $W$ (N) | $Q$ (g/s) |
|---|---|---|---|---|---|---|
| 1 | 0.2000 | 15.00 | 20.0 | 8 | 94.68 | 0.2157 |
| 2 | 0.2055 | 16.52 | 15.6 | 8 | 97.48 | 0.1605 |
| 3 | 0.2049 | 17.15 | 15.0 | 8 | 99.49 | 0.1720 |
| 4 | 0.2017 | 16.97 | 17.7 | 8 | 101.35 | 0.1814 |
| 5 | 0.2079 | 16.25 | 18.0 | 8 | 104.97 | 0.1861 |
| 6 | 0.2088 | 15.93 | 15.7 | 8 | 106.17 | 0.1887 |
| 7 | 0.2004 | 15.06 | 16.1 | 8 | 108.29 | 0.1960 |
| 8 | 0.2025 | 14.86 | 18.3 | 10 | 110.55 | 0.1970 |
| 9 | 0.2040 | 14.27 | 16.9 | 10 | 112.46 | 0.2021 |
| 10 | 0.2068 | 13.89 | 17.3 | 10 | 113.08 | 0.2076 |
| 11 | 0.2098 | 13.14 | 17.4 | 10 | 116.52 | 0.2136 |

## 5. Conclusions

In this paper, the mathematical model of bearing capacity and gas consumption of aerostatic bearings is established using an engineering calculation method, and the rapid calculation of the static performance of aerostatic bearings is realized. The static performance of aerostatic bearings can be solved more accurately using simulation software, so the influence of each structural parameter on static performance can be analyzed using simulation.

Through a trial and error method, we found that the MOPSO algorithm using Latin hypercube initialization is more stable than using random initialization and shows better convergence speed and convergence results in most cases.

Pareto solutions obtained using the MOPSO algorithm using Latin hypercube initialization all meet the two optimization conditions of improving the bearing capacity and reducing gas consumption. The MOPSO algorithm using Latin hypercube initialization offers more abundant and more accurate global optimal solutions. Designers can further filter out the optimal solution by allocating the proportion of bearing capacity and gas consumption and verify the design result using the simulation model. The optimization design method proposed in this paper offers certain practical guiding value for the rapid optimization design of aerostatic bearings.

**Author Contributions:** Data curation, G.Y.; formal analysis, G.Y. and Y.Z.; resources, Y.Z.; validation, G.Y. and Y.Z.; writing—original draft, G.Y.; writing—review and editing, B.Y. and G.L. All authors have read and agreed to the published version of the manuscript.

**Funding:** This project was funded by the Zhejiang Basic Public Welfare Research Program (LGG19E050025).

**Institutional Review Board Statement:** This paper does not involve humans or animals.

**Informed Consent Statement:** This paper does not involve humans or animals.

**Data Availability Statement:** The study did not report any data.

**Acknowledgments:** Thanks for the support of the Zhejiang Basic Public Welfare Research Program (LGG19E050025).

**Conflicts of Interest:** The authors declare no conflict of interest.

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
