# Peer review of "Structural Design of Aerostatic Bearing Based on Multi-Objective Particle Swarm Optimization Algorithm"

_applsci, doi:10.3390/app13053355_

Round 1
Reviewer 1 Report
A very nice paper overall.
Authors developed an engineering calculation method for an aero-static bearing. And in section 3.2, they showed the accuracy of the model by comparing it with the simulation results and the test results from another study. Then, using various optimization algorithms found the best working algorithm for their model. One small comment:
A comparison of the processor solution times of simulation vs developed analytical model and a comparison of solution times of different optimization algorithms used may be helpful.
Reviewer 2 Report
Using Latin hypercube Sampling to modify the particle distribution in the first cycle is not innovative.
It is suggested to conduct a more comprehensive study on similar works in the introduction and examine the strengths and weaknesses of previous works.
Explain the stopping conditions of the algorithm. Why is the algorithm stopped after 60 cycles? What is the justification for convergence?
A change in the Pareto slope can be seen at the end region of the obtained Pareto (in the range of 0.9 to 1). What is the authors' assessment of its cause?
Due to the existence of the random term in the original formulation of the MOPSO Algorithm, it is unlikely that the modification of particles in the first cycle will have a significant effect. It is suggested to use several benchmark problems (for example CEC problems) to prove the efficiency of the proposed method.
It seems that the Diversity Metric should be a number close to one (for better performance). While Figure 14 shows the opposite.
To control the convergence of the two-objective optimization algorithm, it is recommended to find the optimal answer of each objective as a single objective problem.
Information from many references is incomplete (for example, year of publication).
The conclusion section needs to be rewritten.
Round 2
Reviewer 2 Report
The paper is now accepted.